# Unveiling the Protein Components of the Secretory-Venom Gland and Venom of the Scorpion *Centruroides possanii* (Buthidae) through Omic Technologies

**DOI:** 10.3390/toxins15080498

**Published:** 2023-08-09

**Authors:** Patricia Elizabeth García-Villalvazo, Juana María Jiménez-Vargas, Gisela Jareth Lino-López, Erika Patricia Meneses, Manuel de Jesús Bermúdez-Guzmán, Carlos Eduardo Barajas-Saucedo, Iván Delgado Enciso, Lourival Domingos Possani, Laura Leticia Valdez-Velazquez

**Affiliations:** 1Facultad de Ciencias Químicas, Universidad de Colima, Colima 28400, Mexico; pgarcia29@ucol.mx (P.E.G.-V.); jjimenez45@ucol.mx (J.M.J.-V.); carlos_barajas24@ucol.mx (C.E.B.-S.); 2Consejo Nacional de Humanidades, Ciencia y Tecnología (CONAHCYT), Mexico City 03940, Mexico; 3Centro Nacional de Referencia de Control Biológico, Dirección General de Sanidad Vegetal SENASICASADER, Colima 28110, Mexico; giselalino01@gmail.com; 4Departamento de Medicina Molecular y Bioprocesos, Instituto de Biotecnología, Universidad Nacional Autónoma de México, Cuernavaca 62210, Mexico; erika.meneses@ibt.unam.mx; 5Agrícolas y Pecuarias (INIFAP), Instituto Nacional de Investigaciones Forestales, Colima 28110, Mexico; bermudez.manuel@inifap.gob.mx; 6Facultad de Medicina, Universidad de Colima, Colima 28040, Mexico; ivan_delgado_enciso@ucol.mx

**Keywords:** *Centruroides possanii*, next generation sequencing, omics technologies, proteome, scorpion, transcriptome, venom, venom gland

## Abstract

*Centruroides possanii* is a recently discovered species of “striped scorpion” found in Mexico. Certain species of *Centruroides* are known to be toxic to mammals, leading to numerous cases of human intoxications in the country. Venom components are thought to possess therapeutic potential and/or biotechnological applications. Hence, obtaining and analyzing the secretory gland transcriptome and venom proteome of *C. possanii* is relevant, and that is what is described in this communication. Since this is a newly described species, first, its LD_50_ to mice was determined and estimated to be 659 ng/g mouse weight. Using RNA extracted from this species and preparing their corresponding cDNA fragments, a transcriptome analysis was obtained on a Genome Analyzer (Illumina) using the 76-base pair-end sequencing protocol. Via high-throughput sequencing, 19,158,736 reads were obtained and ensembled in 835,204 sequences. Of them, 28,399 transcripts were annotated with Pfam. A total of 244 complete transcripts were identified in the transcriptome of *C. possanii*. Of these, 109 sequences showed identity to toxins that act on ion channels, 47 enzymes, 17 protease inhibitors (PINs), 11 defense peptides (HDPs), and 60 in other components. In addition, a sample of the soluble venom obtained from this scorpion was analyzed using an Orbitrap Velos apparatus, which allowed for identification by liquid chromatography followed by mass spectrometry (LC-MS/MS) of 70 peptides and proteins: 23 toxins, 27 enzymes, 6 PINs, 3 HDPs, and 11 other components. Until now, this work has the highest number of scorpion venom components identified through omics technologies. The main novel findings described here were analyzed in comparison with the known data from the literature, and this process permitted some new insights in this field.

## 1. Introduction

Scorpions are evolutionarily successful arachnids, having remained on the planet throughout environmental changes, mass extinctions, and recent coexistence with humans since their appearance approximately 450 million years ago. Since then, they have adapted to physiological changes and numerous ecological, biochemical, and ethological situations [1,2]. 

Throughout their evolution, they have developed the ability to produce venom, allowing them to hunt down their prey and defend themselves from predators. Human health could be affected by their venom. Of the 2762 described species [3], only 50 species are suspected of causing severe envenoming and deaths to humans [2].

Ancient civilizations have used scorpion venom to treat some illnesses and diseases, even for the stings themselves [4,5]. In recent decades, novel technological developments have allowed for the analysis of venom components, as well as the determination of their chemical structures and functions. In particular, the protein components of scorpion venom have been classified into five families according to their structural domain: toxins, enzymes, host defense peptides (HDPs), protease inhibitors (PINs), and other components. Of these, neurotoxins are the most relevant because they act on sodium channels (NaTx), generating severe symptoms of envenoming and even death to mammalians [6]. These compounds have been identified through studies of the venom-producing gland by transcriptomic analysis and proteomics studies performed on the soluble venom. 

The scorpion *Centruroides possanii* (now and then abbreviated *C. possanii*) was recently described by González-Santillán et al. [7]. It is endemic to the ecologic reserve El Terrero, Minatitlan, Colima, Mexico, and ranges in color from straw yellow to ochraceous orange. It is the smallest described *Centruroides* species [7] (Figure 1). The venomous components of this scorpion have not previously been investigated, and we intend to present such findings. The number of cases of scorpionism causing morbidity in Colima State was 9015 in 2022 [8], and these were caused by only five species, including *C. possanii*. The symptoms can be mild, moderate, or severe, and start with local discomfort in the sting area which progresses to dyspnea, which may cause the victim’s death [9].

Recent studies have permitted researchers to correlate the structures of the peptides and proteins with their functions, enriching the information contained in the database, improving antivenom, and identifying new therapeutic agents and compounds with biotechnological applications. Furthermore, the medium lethal dose of venom was determined. All of this evidence suggests that the scorpion *C. possanii* could be considered one of the most toxic Mexican species.

## 2. Results and Discussion

### 2.1. Determination of LD_50_ of C. possanii Venom

The LD_50_ of *C. possanii* venom has been estimated to be 13.18 ± 0.67 μg/20 g mouse, or 659 ± 33.5 ng/g of mouse weight. The dose was validated using six mice, and death was observed in 50%. Mice injected with this venom presented intoxication signs, such as squealing, jumping, goosebumps, difficulty breathing, and death in a few minutes. The signs which were observed may have been related to neurotoxins present in *C. possanii* venom, which affect excitable cells, as has been described for other scorpions of the same genus, like *C. tecomanus* (10.2 μg/20 g mouse) and *C. limpidus* (15.0 μg/20 g mouse) [10,11]. The scorpion *C. possanii* should be considered of medical importance.

### 2.2. RNA Integrity and Sequenciation Quality

The results obtained by the Bioanalyzer 2100 (Agilent Technologies, Waldbronn, Germany) showed little RNA degradation (Figure 2). The RNA integrity number (RIN) determines the RNA integrity through 18S and 28S peaks observed in the electropherogram. However, in scorpions, there is a dissociation of the 28S rRNA subunits, causing these to migrate on a gel at the same rate as the 18S rRNA subunit, which does not permit the determination of real RIN [12]. Nonetheless, a clearly defined 18S rRNA band in the gel and its peak in the electropherogram were observed, indicating the RNA’s quality and integrity. The RNA concentration of the sample was 104 ng/µL. The Phred quality level of the reads was 30, where the probability of finding an incorrect base is 1 in 1000, with an accuracy of 99.9%.

### 2.3. Bioinformatics

In the transcriptome of *C. possanii*, 19,158,736 reads were obtained with 76 base pairs (bp), as expected. The reads were assembled de novo, and this resulted in 835,204 transcripts. From them, 720,463 were annotated with some biological function. Of this, 434,492 were putative venom transcripts of the scorpion. The Pfam annotation resulted in 28,399 transcripts. These were classified into 56 Pfam scorpion domains, of which 50 were specific to Pfam scorpion venom domains. Table 1 shows the results of the assembly and annotation.

### 2.4. Venom Components Found in the Secretory Venom Gland of C. possanii

A database was generated for identification of the components which were found. We downloaded another database of scorpion venom components and other venomous species, such as spiders; sea cone snail anemones; and snakes, from UniProt, Animal Toxin Annotation Project, ArachnoServer, PDB, NCBI, and ScorpKTx. With all of the above information, a single local database was developed and 434,606 unique venom sequences were obtained, of which 162,285 were toxins, 129,176 enzymes, 7757 host defense peptides, 39,895 protease inhibitors, and 48,316 classified as other peptides. The curing of the sequences was performed by means of the BLASTp algorithm, using a transdecoder file with the local database as the first step for obtaining the list of transcripts to cure. Through transcriptomic analysis, the sequences with identity to venom components were identified, and the enzymes were the most diverse group (Table 2). However, the sodium channel-acting toxins (NaTx) were the most abundant. The TMP values indicate the relative abundance of the transcripts, shown in Appendix A.

#### 2.4.1. Ion Channel-Acting Toxins

##### Sodium Channel-Acting Toxins

Previous reports of transcriptomes in scorpions with different levels of toxicity to mammalian species have shown that toxins acting on sodium channels are the most abundant peptides in the venom of these arachnids, especially those of the *Buthidae* family [10]. This is also the case for *C. possanii*, in which 218 (110 complete) different coding transcripts surmised to affect NaTxs were identified (Appendix A). The transcripts were named CpoNaTx. 

*α-NaTx*. Analysis of the results showed 39 coding transcripts for α-NaTx, 21 of which were complete. The CpoNaTAlp10 sequence showed 69% identity with the peptide CsE5 (UniProt P46066) from the venom of *C. sculpturatus*, which is toxic to vertebrates and invertebrates, but more toxic to vertebrates. The toxicity of this protein to mammalians may be due to the presence of lysine 32 and 75 (Figure 3), as well as in beta toxins AaH2 from the scorpion *Androctonus australis* and Css2 from *C. suffusus*. The capacity of CsE5 to block neuronal transmission and induce spontaneous rhythmic muscle contractions is due to structure–function relationships. In insects, the toxicity is related to the additional three C-terminal residues (N^81^, P^82^, A^83^) [13]. CpoNaTAlp10 also has 67% identity with the toxin Pg8 (UniProt B7SNV8), which is the most toxic component of the venom of the South African scorpion *Parabuthus granulatus* [14] (Figure 3). Toxin Pg8 acts on site 3 of sodium channels, slowing the inactivation process [15]. 

The sequence CpoNaTAlp05 shows an identity of 57% and 35% with the CvIV4-like toxin from *C. sculpturatus* and Cn12 from the *C. noxius* scorpion, respectively. These are alpha toxins, but they show tridimensional conformation similar to beta toxins. Experimental data show that it behaves like an alpha-type toxin by inducing inactivation of the sodium channel [6]. The presence of CpoNaTAlp05 was also identified in the soluble venom, from which three unique peptides were found (Appendix A).

*β-NaTx.* Fifty complete sequences of coding transcripts in the transcriptome and fourteen proteins in proteome analyses were identified. These peptides are the most abundant in the venom of scorpions of the *Centruroides* genus, and are the main toxins responsible for causing the symptoms and signs of envenoming in mammalians [16]. *C. hirsutipalpus*, with 77 β-NaTxs, is thus far the scorpion described as having the highest number of transcripts assumed to code for sodium channel toxins affecting mammals [17], whereas *C. limpidus* contains 59 transcripts [10], *Tityus obscurus* 48 [18], and *C. hentzi* 36 transcripts [19].

CpoNatBet09, CpoNatBet14, and CpoNaTBet35 were the most abundant both in the transcriptome and the proteome (Appendix A). The sequence CpoNatBet09 was 100% identical to the Cll2b toxin (UniProt P59899) from the *C. limpidus;* it presented identities of 89% and 86% with the toxins Css4 (UniProt P60266; *C. suffusus*) and Cn2 (UniProt P01495; *C. noxius*), respectively (Figure 4). These toxins are highly represented in their venoms, and present regions that are conserved in their amino acid sequences and are essential for their activity on sodium channels (L^39^, N^42^, Y^44^, R^47^, E^48^, Q^52^, W^78^) [20,21]. In the case of the Cn2 toxin, it represents 6.8% of the total venom, with an LD_50_ of 0.25 μg/20 g of the mouse to which it was administered intraperitoneally [22,23]. It also showed affinity to Nav 1.6 channels [24]. The transcript CpoNaTBet09, which codes for a peptide showing high sequence similarity or even complete identity to the *C. limpidus* toxin Cll2b, is certainly a component that plays an important role in the intoxication process.

##### Potassium Channel-Acting Toxins (KTx)

In the transcriptome of *C. possanii*, we identified 38 complete sequences which coded for toxins that act on potassium channels. Of these, 20 sequences were classified as α-KTx, one *β*-KTx, five γ-KTx, and twelve δ-KTx (Appendix A). The proteomic analysis permitted the identification of five of these sequences: one *β*-KTx, three α-KTx, and one γ-KTx (Appendix A). This indicates that the transcriptome of *C. possanii* has the highest number of KTx-similar transcripts reported thus far, when compared with 26 of *C. limpidus* [10] and 21 of *C. hirsutipalpus* [17].

*α-KTx*. In the transcriptome of *C. possanii*, one coding transcript with an identity to α-KTx, called CpoKTxAlp04, was identified, which showed 100% identity with cobatoxin 1 (UniProt O46028) of the scorpion *C. noxius* and 96.9% identity with the KTx 10.4 toxin (UniProt C0HJW2) from *C. tecomanus* (Figure 5). These toxins consist of a polypeptide chain of 32 amino acids stabilized with 3 disulfide bridges. In their mature sequences, five conserved residues are present: K^49^, I^51^, N^52^, K^56^, and Y^58^. These are believed to be important to the interaction with the channel Kv1.1 channels [25].

*β-KTx*. The CpoKTxBet01 sequence was identified in the proteomic analysis, and two unique peptides were found (Appendix A). It showed similarity with the TdiKIK-like toxin (NCBI XP_023211330) of *C. sculpturatus*.

*γ-KTx*. Twenty-two coding transcripts were found with identity to gamma toxins. Five coding transcripts were complete (Appendix A). The CpoKTxGam01 coding transcript showed identities of 55% and 52% with CnErg1 (UniProt Q86QT3, *C. noxius*), CeErg1 (UniProt Q86QV6, *C. elegans*), and CmERG1 (UniProt C0HLM3, *C. margaritatus*) toxins, respectively (Figure 6). These toxins contain residue of lysine in position 34 (K^34^) in their sequences, which could be the principal interaction mode with the potassium channel [26]. The CnErg1 toxin blocks the Kv11.1 and 11.3 channels in a closed state in humans and rats [27], while the CmERG1 toxin only blocks the Kv11.1 channel [28]. The CpoKTxGam05_01 sequence was identified in the proteome, where two unique peptides were identified (Appendix A). This peptide showed 100% identity with the CllErgTx2 toxin (UniProt Q86QU9) from *C. limpidus*.

*δ-KTx*. The peptides of this subfamily have a Kunitz-type structural folding with protease activity and Kv1.3 potassium channel blockers [29]. According to this classification, ninety-four coding transcripts were identified. Still, only 12 sequences were complete. Of these, the coding transcripts (CpoKTxDel01-10) showed identity with the BmKTT-2 toxin (UniProt P0DJ50) from the scorpion *Mesobuthus martensii*, presenting an identical pattern of eight cysteines in their sequence, and were classified in group 1. Two coding transcripts (CpoKTxDel11 and 12) with sequences like the BmKTT-1 toxin (UniProt P0DJ49) from *Mesobuthus martensii* were found and categorized in group 2. The CpoKTxDel03 coding transcript shows 86% identity with the BmKTT-2 toxin and 55% and 50% identity with the Kunitz-type carboxypeptidase inhibitor Kci-1 (GenBank AIX8708, *Androctonus bicolor*) and Conotoxin precursor conkunitzin, partial (GenBank Uma82726, *Conus ebraeus*), respectively (Figure 7). The BmKTT-2 toxin inhibits trypsin and chymotrypsin and is a potent inhibitor of human plasmin, suggesting it may have good stability for oral drug delivery [30]. In contrast, the putative peptides of group 2, i.e., the CpoKTxDel11 and 12 showed 86% and 56% identity with peptides of Kunitz-type serine protease inhibitor LmKTT-1a-like from *C. sculpturatus* (NCBI XP_023242637) and *Lychas mucronatus* (UniProt P0DJ46), respectively, as well as 50% identity with venom protein VP5 (GenBank ALX72370) of *Odontobuthus doriae* [31] (Figure 8). Although the LmKTT-1a toxin completely inhibits trypsin activity, it does not affect chymotrypsin or elastase and blocks potassium currents in the Kv1.3 channel [29,32].

#### 2.4.2. Enzymes

Enzymes are a group of diverse components identified in the transcriptome of *C. possanii* and other species, like *C. hirsutipalpus* [17] and *C. limpidus* [10]. This study reports the presence of 915 coding transcripts showing identity to enzymes. They can be classified into 12 subfamilies: cysteine proteases (CiP), serine proteases (SeP), metalloproteases (MtP), monooxygenases (Mon), hyaluronidases (Hya), phosphodiesterases (PDE), phospholipase A1 (PLA1), phospholipase A2 (PLA2), phospholipase B (PLB), 5′-nucleotidases (5′-NT), phospholipase C (PLC), and phospholipase D (PLD). Here, 47 transcripts were identified, coding for 9 of the 12 subfamilies (Appendix A).

*PLA1*. Four complete coding transcripts (CpoEnzPA101-04) with identity to phospholipases A1 were identified in the transcriptome, and three sequences (CpoEnzPA101, 03 and 04) in the proteomic analysis (Appendix A). Enzymes from this group have been reported in venom gland transcriptomes of non-toxic scorpions, such as *Hadrurus spadix* [33] and *C. hentzi* [19]. Unfortunately, none of these have experimental evidence of their function or the presence of the protein in the venom. This is the first report that has determined the presence of PLA1 in a transcriptomic and proteomic analysis of scorpion species known to be toxic to mammalians.

*PLA2*. Phospholipases A2 (PLA2) found in scorpion venom presented histidine (H) and aspartic acid (D) residues conserved in the active site, according to the following proteins, the function of which was already experimentally proven: Plaiodactylipin (UniProt Q6PXP0) from *Anuroctonus phaiodactylus*, HgPLA2 (UniProt P0C8L9) from *Hoffmannihadrurus gertschi*, HfPLA2 (UniProt Q3YAU5) from *Chersonesometrus fulvipes*, Imperatoxin-1 (UniProt P59888) from *Pandinus imperator*, Heteromtoxin (UniProt P0DMI6) *Heterometrus laoticus*, and MtsPLA2 (UniProt Q6T178) from *Hottentotta tamulus*. Four coding transcripts with identity to PLA2 (CpoEnzPA201-04) were identified in this transcriptome (Appendix A) and the CpoEnzPA203 sequence in the proteomic analysis (Appendix A). PLA2 hydrolyzes phospholipids, modifying the structure of the membranes and performing pre-digestive functions, and could be considered a spreading factor. They are also the most widely distributed enzyme in animal venoms [34].

*PLB*. The complete transcript was named CpoEnzPLB01 and presented 98.2% identity with the phospholipase B-like 1 (UniProt Q6P4A8) from the scorpion *C. sculpturatus*. The CpoEnzPLB01 coding transcript showed identity with the Phospholipases B-like (UniProt A0A1W7R9X8) from *Hadrurus spadix* and (UniProt A0A2I9LPG8) from *C. hentzi*. The CpoEnzPLB01 complete coding transcript presented the Pfam PF04916 (Phospholipase B) domain and the conserved Phospholipase B domain, according to CDD/Sparcle [35]. Six unique peptides of CpoEnzPLB01 coding transcripts were found in the proteome (Appendix A). The exact role of this enzyme in scorpion venom has not yet been described, but in snake venom, it is known to act as a potent hemolytic agent [36].

*Hya*. One coding transcript (CpoEnzHya01) with an identity to hyaluronidase was identified in the transcriptome and in the venom from *C. possanii*. Its sequence presented 98.9%, 75.9%, and 73.3% identity with the hyaluronidase-1-like (NCBI XP_023226974.1) from *C. sculpturatus*, hyaluronidase 1 (UniProt P85841) from *Tityus serrulatus*, and (UniProt P86100) from the scorpion *Mesobuthus martensii*, respectively. These sequences contain an aspartic acid at position 131 (D^131^), corresponding to the active site. Its function in the venom is well documented. It degrades the hyaluronic acid of the extracellular matrix, thus allowing for the diffusion of other venom components, such as toxins, increasing its potency and damaging the local tissue at the sting site [37].

*Mtp*. Sixteen complete transcripts were identified as coding for metalloproteinases, including antarease-like, astacin-like, and angiotensin-converting enzyme-like. The CpoEnzMtP06 coding transcript has 91.7% identity with the antarease-like TtrivMP_A (NCBI XP_023228637.1) from *C. sculpturatus.* These two enzymes contain 505 amino acids and present the same pattern of cysteines, forming nine disulfide bridges. The region with metalloprotease activity comprises amino acids 181 to 405; it has been proposed that residues 338 to 348 constitute the active site of the enzyme. Eleven metalloproteases of the transcriptome also were found in the proteome (Appendix A).

On the other hand, a sequence of 606 amino acids with identity to angiotensin-converting enzyme (ACE)-type protease was found. In mammalians, this protein plays a role in the renin-angiotensin system, which regulates blood pressure and chloride retention via the kidney. The ACE protein converts angiotensin I to angiotensin II, its active form, resulting in increased vasoconstrictor activity and causing an elevation of peripheral vascular resistance, therefore leading to an increase in blood pressure [38]. ACE-type proteins have been found in some sea cone and wasp species. In scorpions, they have been reported in transcriptomic analyses of *Hottentotta judaicus* [39], *Tityus bahiensis* [40], *T. stigmurus* [41], and *T. serrulatus* [42]. ACE-like protein, one of the components of scorpion venom, may contribute to the arterial hypertension observed in human victims during envenomation caused by the sting of this arachnid [40]. The CpoEnzMtp08 coding transcript showed identity mainly with two ACE-like proteins from *C. sculpturatus* (NCBI XP_023209358.1) and *Tityus serrulatus* (UniProt JAW07033.1). These sequences present the conserved HHEXXH motif of metallopeptidases as well as acetylated and deamidated regions.

*CiP*. Two coding transcripts with identity to cysteine proteases were identified in the transcriptome (Appendix A), and one sequence in the proteome (Appendix A). Cysteine proteases have not been reported in species of the *Centruroides* genus. But they have been identified in the transcriptome of *Hadrurus spadix* [33], *T. obscurus*, *T. serrulatus* [18], and *T. bahiensis* [40]. The coding transcript CpoEnzCiP01 has 97.2% identity with cathepsin L-like precursor (NCBI XP_023238299.1) from the scorpion *C. sculpturatus*. This sequence presents its cysteine protease domain in the 236–453 position of the amino acid sequence. CpoEnzCiP01 also showed 83.4% identity with the cathepsin F-like (UniProt U6JPB2) from the scorpion *Tityus serrulatus*, and its cysteine protease domain was found in the 228–441 positions of the sequence. The cysteine proteases are large proteins, and the sequences comprise 446 (CpoEnzCiP01 and *T. serrulatus*) and 454 (*C. sculpturatus*) amino acids. The cysteine protease activity is thought to be at the carboxyl terminal. This family of proteins has the cysteine–histidine–asparagine catalytic triad at the active site (C^389^, H^396^, and N^422^) (Appendix A).

*Mon*. Two coding transcripts with identity to monooxygenase were identified in the transcriptome and the proteome: CpoEnzMon01 and CpoEnzMon02 (Appendix A). The CpoEnzMon01 transcript showed identity with the monooxygenase sequences (NCBI XP_023239442.1) from *C. sculpturatus* (97.5%) and *C. hentzi* (NCBI A0A2I9LP74; 92.3%). One of the main activities of the monooxygenases is to carry out post-translational modifications to some toxins, particularly through amidation of the C-terminus (α-amidation), which is necessary for the toxins to be functional [43].

*SeP*. Fourteen transcripts coding for serine proteases were identified in the transcriptome, and four in the proteome. The CpoEnzSeP04 coding transcript had 99.4% identity with the serine proteinase stubble-like (NCBI XP_023209481.1) from *C. sculpturatus.* Both sequences contained 360 amino acids and the same cysteine pattern (Appendix A). These enzymes presented a region with trypsin-like activity in the 116–355 positions of the amino acids. Their active sites were formed by the H^163^, D^213^, and S^311^, while the substrate binding sites comprised D^305^, S^331^, and G^333^ residues. Serin proteases are abundant enzymes of the venom of viperids [44]. Although their functions are diverse, they generally have hemotoxic activity, acting as anticoagulants or procoagulants [44].

*PDE*. Twenty-three sequences showed identity with phosphodiesterases; three of them were complete sequences and identified as CpoEnzPDE01-CpoEnzPDE03 in the transcriptome and proteome results. Phosphodiesterases have been reported at the transcriptomic level of the scorpion venom gland [45], in the venom of snakes [46], and in *Loxosceles* spiders, where it is the component responsible for causing dermo-necrosis and systemic effects [47]. The CpoEnzPDE01 coding transcript shows 97.6% identity with the phosphodiesterase-type sphingomyelin (NCBI XP_023216777.1) from the scorpion *C. sculpturatus* (Appendix A). This was the first report demonstrating the presence of phosphodiesterases in scorpion venom. The great diversity and abundance of enzymes in the venom could contribute to the effects of intoxication observed in their prey and act as spreading factors, allowing for the diffusion of the venom components throughout the body of the victim. They also could be activators of other venom components due to their preservative or multifunctional qualities [45].

#### 2.4.3. Host Defense Peptides (HDPs)

##### Defensins

Ten complete coding transcripts were identified with identity to defensins (Appendix A), and two of them were found in the proteome (CpoHDPDef07 and CpoHDPDef10; Appendix A). Defensins are small peptides (less than 50 amino acids) containing cysteines. They were found in insects, mollusks, and arachnids [48]. The transcriptomic analysis of venom secretory glands from diverse scorpion families has allowed for the description of coding transcripts with identity to defensin-like sequences, which are more abundant in species of scorpions that are non-toxic to mammalians [10].

The CpoHDPDef01 coding transcript has 83.3%, 75%, and 75% identity with dual-function peptides such as Bmkdfsin3 (UniProt A0A384E0Y8) from scorpions *Buthus martensii*; 4 kDa defensin (UniProt P56686) from *Androctonus australis;* and 4 kDa defensin (UniProt P41965) from *Leiurus hebraeus*, respectively (Figure 9). These sequences present a conserved cysteine pattern, forming three disulfide bridges (4–25, 11–33, 15–35). 

The Bmkdfsin3 toxin acts on the Kv1.1, Kv1.2, and Kv1.3 channels, inhibiting their current, and has potent antibacterial activity on Gram-positive bacteria such as *Staphylococcus aureus* (2.5 μM), *Micrococcus luteus* (2.5 μM), and *Staphylococcus epidermidis* (1.25 μM) [49]. The 4 KDa defensin from *L. hebraeus* also shows activity against *Bacillus subtilis*, although it is not considered a human pathogen; it could contaminate food. None of the above peptides show hemolytic activity [50]. Therefore, they could be potential candidates for the development of new antimicrobial agents.

##### Non-Disulfide-Bridged Peptide (NDBP)

These peptides do not contain cysteines, but have a high diversity of bioactivities, such as bradykinin-potentiating [51], antimicrobial [52], anticancer [53], and cytolytic and hemolytic activity [54]. The NDBPs were classified into five groups according to their pharmacological action, sequence similarity, and length [54,55]. This is the first report demonstrating the presence of NDBP-3 (CpoHDPND301) in the venom of scorpions of the *Centruroides* genus. This peptide presented a 74% identity with the NDBP 4.23-like (UniProt Q5G8B5) coding transcript of the scorpion *Tityus costatus*.

#### 2.4.4. Peptides Inhibitors (PINs)

Seventeen complete coding transcripts with identity to protease-inhibitory peptides (PINs) were identified in the transcriptome. These were classified into three subfamilies: the cysteine-rich trypsin inhibitor domain (TIL domain), the serine protease inhibitor domain (SERPIN), and those with a Kunitz-type domain. The TIL protease members include anticoagulant proteins, elastase inhibitor, thrombin, trypsin, and chymotrypsin [56]. This domain contains ten cysteines forming five disulfide bridges (1–7, 2–6, 3–5, 4–10, and 8–9) in a sequence of 60 amino acids. The active site of these inhibitors is in the loop connecting beta sheets 1 and 2, located between two disulfide bridges [57]. In the transcriptome of *C. possanii*, eleven coding transcripts were identified with identity to TIL proteases. The CpoPInTIL07 coding transcript shows 76.3% identity with the VP5.1 peptide (GenBank ABY26681) from the scorpion *Lychas mucronatus*, with cysteines in the same positions and a chain length of 60 amino acids in the mature peptide.

Five complete and eight partial coding transcripts with identity to SERPIN-type proteases were identified (Appendix A). This family is involved in inflammatory responses, complementary activation, blood coagulation, and mutations associated with emphysema, cirrhosis, and dementia [58].

The CpoPInKun01 coding transcript shows 80% identity with isoinhibitor K-like (NCBI XP_0232117496) from the scorpion *C. sculpturatus.* Both sequences contain seven cysteines. Finally, fragments of two TIL and four SERPIN-type proteases were identified in the proteomic analysis (Appendix A).

#### 2.4.5. Other Components

This category comprises various proteins, including insulin-like growth-factor-binding proteins (IGFBPs); members of the cysteine-rich secretory protein group, antigen 5, and pathogenesis-related 1 proteins (CAP superfamily); La1-like peptides with von Willebrand factor-containing proteins; and orphan or undefined proteins. The common characteristic among these groups is that their functions as either transcripts or venom components remain unclear. Seventy-four coding transcripts were classified within this category, with sixty complete sequences (Appendix A).

##### Insulin-like Growth Factor-Binding Proteins (IGFBP)

Twenty-seven coding transcripts were identified in which a conserved IGFBP domain was found (Appendix A). One of them was also identified in the proteome (Appendix A). In this report, IGFBPs were the most abundant group in the “Other components” category. Although their function in venom is unknown, IGFBPs have been reported in previous transcriptomic studies [17,19,32,33,40,59], as well as in proteomes [17,33,40]. The CpoOthIGF02 coding transcript had an identity of 98% with the venom protein 302-like (NCBI XP_023234366) of *C. sculpturatus* and a 57% identity with the AbOp-5 peptide (GenBank AIX87724) from *Androctonus bicolor*. 

##### Cysteine-Rich Secretory Proteins, Antigen 5, and Pathogenesis-Related 1 Proteins (CAP)

Protein groups belonging to this superfamily have been documented in arthropods, mammalians [60], and reptiles [61]. These secreted proteins have a generalized extracellular function, acting in an endocrine or paracrine manner. They are involved in various processes, including regulating branching morphogenesis and the extracellular matrix, potentially as proteases or protease inhibitors; modulating ion channels in fertility as tumor suppressors; and during fertilization in cell–cell adhesion [62]. In scorpions, their function in the venom remains unknown. However, about 119 coding transcripts have been reported in transcriptomic analyses, and their presence in venom has been verified in species of the Buthidae, Vaejovidae, Caraboctonidae, Superstitionidae, and Euscorpiidae families [18,33,63]. Seven coding transcripts with identity to CAP proteins were identified in this transcriptomic analysis, and three of them were found in the proteome (Appendix A).

##### Orphan or Undefined Coding Transcripts 

These components with no defined structural domain and no known function have been found in *Centruroides* scorpions: four coding transcripts and three peptides in *C. hentzi* [19], and three peptides in *C. limpidus* [10]. We identified nineteen coding transcripts with identity to orphan peptides, and five peptides were also found. This is the highest number of coding transcripts with identity to orphan peptides which has been reported in scorpions of the *Centruroides* genus to date, and the second most abundant group within this category according to transcriptome analysis. Given the information above, this group of peptides will certainly be studied further in the near future.

##### La1-like Peptides with Single-Domain von Willebrand Factor

The La1-like peptides are named after the first peptide to be identified in the venom of the *Liocheles australiase* scorpion of the family Hormuridae. These peptides present the structural domain of von Willebrand factor type C (SVWC) and contain eight cysteines within their amino acid sequences. Generally, their function is to respond to environmental challenges such as bacterial infections or antiviral immunity [64]. In the analysis of the transcriptome of venom-secreting glands of the scorpion *C. possanii*, seven coding transcripts were found with identity to proteins with the SVWC domain (Appendix A). In the proteome, one protein was identified (Appendix A). 

### 2.5. Venom Proteomic Components of C. possanii Scorpion

#### 2.5.1. Peptide Mass Fingerprinting

A fraction of the soluble venom from the scorpion *C. possanii* was analyzed using a mass spectrometer for molecular mass identification. The system yielded 180 unique individual masses with molecular weights of 800 to 17,000 Daltons (Da), as shown in Appendix A. The most abundant components (39 peptides) presented molecular weights between 7001 and 8000 Da, attributed to toxins acting on sodium channels. This was related to the number of transcripts and peptides identified in the transcriptome and proteome, respectively, of the *C. possanii* scorpion. The lowest number of compounds (8) was found in the 800–1000 Da range. Additionally, 30 components were identified in the range of 1001 to 2000 Da, 23 components in 2001–3000 Da, 18 in the range of 3001–4000 Da, 24 in 4001–5000 Da, 10 in 5001–6000 Da, 19 in the range of 6001–7000, and only 9 in the range of 8001 to 17000 Da. Table 3 shows a summary of the results regarding the number of peptides and types of proteins identified in the venom of the *C. possanii* scorpion.

#### 2.5.2. Proteomics: Peptides and Proteins Identified in the *C. possanii* Scorpion Venom

In the LC-MS/MS, 50,873 spectra were analyzed, identifying 70 peptides and proteins. Between them were 23 toxins assumed to act on ionic channels. These consisted of four α-NaTx, fourteen β-NaTx, three α-KTx, one β-KTx, and one γ-KTx (Appendix A). Also, 27 enzymes were identified. These were classified into nine subfamilies: three phospholipases A1, one phospholipase A2, one phospholipase B, eleven metalloproteases, four serine proteases, one hyaluronidase, one cysteine protease, three phosphodiesterases, and two monooxygenases. Six protease-inhibitor peptides were identified: four of the SERPIN type and two of the TIL domain. The host defense peptide family was the least abundant, represented by two defensins and one NDBP-3. Finally, in the category of other components, four CAPs, one IGFBP, one La1-like, and five undefined peptides were identified (Table 3).

### 2.6. Correlation between Transcriptomics and Proteomics Results

The transcriptomic and proteomic analysis allowed for the identification of the five families of venom components, classified according to their structural domain: toxins, enzymes, PINs, HDPs, and other components. The most diverse family identified in the transcriptome was the toxins, with 109 members, followed by 60 in the group of other components, 47 enzymes, 17 PINs, and 11 HDPs. Instead, in the proteome, the enzymes were the most diverse, with 27 identified proteins, then the toxins with 23, followed by 11 peptides and proteins in the group of other components, 6 PINs, and 3 HDPs. The diversity pattern by family was very similar in the transcriptome and the proteome, the toxin family being the most diverse. This also supports the idea of recognizing *C. possanii* as a toxic species for mammals. Enzymes were greatly diverse in the proteome and induce post-translational modifications of the toxins, allowing for their activation. In addition, enzymes function as dispersal factors, allowing other venom components, such as toxins, to reach their sites of action. As in other species of the genus *Centruroides*, toxins and enzymes are well represented in the transcriptome and proteome of *C. possanii*, while the PIN and HDP families are less diverse in scorpions which are toxic to mammals.

The last published transcriptomes and proteomes for a scorpion of the genus *Centruroides* were released by Cid-Uribe et al. [10] and Valdez-Velázquez et al. [17]. In these, 192 coding transcripts and 46 identified proteins and peptides were reported in *C. limpidus* [10], in addition to 147 identified coding transcripts and 77 proteins and peptides in the venom of *C. hirsutipalpus* [17]. The same sequencing technology and methodology were used in both studies and for *C. possanii*. The successful identification of a large number of coding transcripts was due to the use of the reference database. This local database integrates identified coding transcripts, peptides, and protein sequences downloaded from various public databases, and includes articles on both terrestrial and marine venomous species, not just scorpions. Therefore, because a large amount of reference data was available, it was possible to identify a greater quantity of coding transcripts, peptides, and proteins. In addition to the above, the quality of the RNA, as well as the assembly and precise annotation of the transcriptome, allowed for the systematic curation of sequences that culminated in their analysis and successful identification.

## 3. Conclusions

A total of 244 complete transcripts were identified, coding for the five families of scorpion venom components: 109 toxins, 47 enzymes, 11 host defense peptides (HDPs), 17 protease inhibitors (PINs), and 60 in the category of other components. Transcriptomic and proteomic analyses of the *C. possanii* scorpion concluded that the peptides expected to act on sodium channels were the most abundant components of the soluble venom. The relative abundance of the CpoNaTBet35 peptide was 11.63%, whereas the peptide CpoNaTBeT09 component constituted 3.5% in the proteome. The venomous glands and the soluble venom of *C. possanii* showed the presence of enzymes such as phospholipase A1, phospholipase A2, phospholipase B, metalloprotease, serine protease, hyaluronidase, cysteine protease, phosphodiesterase, and monooxygenase. In this paper, three components (phospholipases A1, NDBP3, and phosphodiesterases) are described for the first time in the literature regarding scorpion species of medical importance. The *C. possanii* scorpion should be considered as a medically importance species due to the large number of toxic components in the venom and the confirmation of the LD_50_.

## 4. Materials and Methods

### 4.1. Collection of Scorpions

*C. possanii* were collected at night using UV light in El Terrero, an area of Cerro Grande Mountain, under the collection permit SEMARNAT SGPA/DGVS/02139/22, which was granted to our group. This geographical elevation is located within the Sierra de Manantlan Biosphere Reserve in the municipality of Minatitlan, Colima (latitude 19°26′40.0″ N; longitude 103°57′02.5″ W), where the discovery of this species occurred and it was described by González-Santillán et al. (2019) [7]. Ten scorpions were collected, carried to the Biological Products Laboratory of the University, and milked, applying an electric stimulation of 10 mA. They were kept in captivity at room temperature with permanent water access and fed with mealworms (*Tenebrio molitor*) or crickets (*Acheta domesticus*). 

### 4.2. Determination of the Medium Lethal Dose (LD_50_) of the Scorpion C. possanii

Twelve male and female mice of the BALB/c strain (25–30 g) of three months of age, obtained from the animal house of the School of Medicine of the University of Colima, were injected intraperitoneally with soluble venom of the *C. possanii* scorpion, adjusting the dosage according to the weight of each mouse. The dose injected into the first mouse was 15 μg/20 g mouse. Six mice were used to determine the dose, and another six mice were used to confirm it. The mice were maintained with humidity and a constant room temperature, with a 12 h dark–light cycle and food and water ad libitum, and were monitored for 24 h after the venom-injected assay. The study was performed with authorization of the University of Colima Ethics Committee (ethical approval code: FCQUCOL-CEIUEA-1). The determination of the LD_50_ was performed using the Up and Down method reported by Dixon (1965) [65].

### 4.3. RNA Isolation, Sequencing and Reads Quality

First, the scorpions were milked by electric stimulation to obtain the venom, and after fourteen days of collection, the telsons from two males and two females of *C. possanii* were dissected under RNAse-free conditions. The SV Total RNA Isolation System kit (Promega, Madison, WI, USA) was used for RNA extraction. The telsons were placed in a microtube with lysis buffer for maceration with a Kontes pestle. The sample was diluted with RNA dilution buffer and incubated at 70 °C for three minutes. Cell debris was separated by centrifugation, and the supernatant was mixed with ethanol at 95%. The product was centrifuged in a column provided by the kit. The column contents were washed with RNA wash solution, and the RNA was eluted in nuclease-free water. RNA was quantified on a Nanodrop, and its integrity was confirmed on the Agilent 2100 Bioanalyzer. Using the Illumina kit, the cDNA library was constructed starting from the total RNA. 

Automated DNA sequencing was performed at the Bioinformatics and Sequencing Unit at the Biotechnology Institute of UNAM in Cuernavaca, Morelos, Mexico. The base pairs of the cDNA fragments were sequenced on a Genome Analyzer (Illumina) using the 76-base pair-end sequencing protocol. After trimming the adapter, the quality of the raw reads was evaluated with the FastQC program, v0.11.8. 

### 4.4. Bioinformatics: Assembly, Annotation and Data Mining

The transcriptome assembly was performed with Trinity version 20.3 software [66,67], and the functional annotation of transcripts was performed using the Trinotate v3.2.1 pipeline (https://github.com/Trinotate/Trinotate/wiki (accessed on 1 July 2020) [68]. The BLAST program was used for the identification of the sequences, and thousands of sequences of proteins contained in the venoms database of UniProt’s Animal Toxin Annotation Project (https://www.uniprot.org/help/Toxins#:~:text=The%20animal%20toxin%20annotation%20project,a%20few%20fish%20and%20platypuses (accessed on 1 July 2020)), ArachnoServer (http://www.arachnoserver.org/ (accessed on 1 July 2020)), PDB, and NCBI were downloaded. Then, a local base made up of sequences extracted from publications of transcriptomes and proteomes from 2007 to 2020 was generated. SignalP Server version 5.0 or 6.0 (https://services.healthtech.dtu.dk/services/SignalP (accessed on 1 July 2020)) allowed for the identification of signal peptides, ProP 1.0 Server (http://www.cbs.dtu.dk/services/ProP/ (accessed on 1 July 2020)) for the identification of propeptides, and DeepLoc 1.0 for determining the subcellular location (http://www.cbs.dtu.dk/services/DeepLoc/ (accessed on 1 July 2020)). The conserved domains were determined with CDD/SPARCLE (https://www.ncbi.nlm.nih.gov/Structure/cdd/wrpsb.cgi (accessed on 1 July 2020 to February 2023)). Multiple sequence alignments were generated with MAFFT, version 7.0 (https://www.ebi.ac.uk/Tools/msa/mafft/ (accessed on 1 July 2020 to 1 February 2023)). The coding transcripts reported in the following section were classified as complete sequences when they contained all amino acids corresponding to the mature peptide. Instead, those sequences with at least 50% of the amino acids of the mature peptide were considered to be partial. The Salmon tool, v1.9.0, (https://github.com/COMBINE-lab/Salmon (accessed on 1 February 2023)) was employed to estimate transcript-level abundance from RNA-seq data using the database of coding transcripts and the raw transcriptome database [69]. The relative abundance of the transcripts was expressed in units of transcripts per million (TPM).

### 4.5. Soluble Venom Collection and Quantification

Scorpions of the species *C. possanii* were milked by electric stimulation to obtain the venom. After adding distilled water, the sample was centrifugated and the supernatant, called soluble venom, was collected. The determination of the protein concentration in the venom was estimated by absorbance at 280 nm using an Evolution 300 UV-VIS spectrophotometer (Thermo Scientific, Madison, WA, USA), assuming that one unit of absorbance was equal to 1 mg/mL of protein. 

### 4.6. Fingerprinting Mass Analysis of Soluble Venom

Five micrograms of soluble venom were applied in an LC-MS system composed of an HPLC nanoflow (Dyonex 3000, Thermo Scientific, San Jose, CA, USA and a mass spectrometer (Velos-Orbitrap, from Thermo Scientific, San Jose, CA, USA). The sample was fractionated online and ionized by a nanospray ion source. For venom fractionation, the liquid chromatography system was stabilized by 95% solvent A (0.01% formic acid in water) using a homemade reverse phase column (10 cm in length, filled with Jupiter 4 μm and Proteo 90 Å resin (Phenomenex, Torrance, CA, USA) for 20 min [14]. Then, the linear gradient was set to pumping solvent B (0.01% formic acid in acetonitrile) from 5 to 80% for 280 min at a 300 nL/min flow rate. All data on the spectra were acquired, and the raw files were extracted. Components with molecular masses lower and larger than 3 kDa were reported as monoisotopic and average masses, respectively. The allowed error between the theoretical and experimental molecular weight was 1 Da due to the sensitivity of the equipment.

### 4.7. Proteomic Analysis of Venom

One hundred micrograms of soluble venom were reduced with 50 mM dithiothreitol (Sigma Aldrich, Saint Louis, MO, USA) at 56 °C for 30 min, then alkylated with 10 mM iodoacetamide (Sigma Aldrich, Saint Louis, MO, USA) and incubated at room temperature for 30 min. After this treatment, the sample was desalted with C_18_ ZipTips (Millipore; Billerica, MA, USA) and digested with trypsin (Promega, Madison, WI, USA) in 50 mM ammonium bicarbonate buffer. Two micrograms of the digested product were applied into the LC-MS system, which was composed of a Dionex UltiMate 300 nanoflow HPLC (Thermo Scientific, San Jose, CA, USA) with a nanoelectrospray ion source. The compounds were separated in the C18 column (Jupiter 4 μm Proteo 90 Å resin), employing a gradient from 10% to 80% solvent B in 250 min. The peptidic fragments were analyzed in positive ion mode. Mass spectra were acquired in a full-scan *m*/*z* range between 350 and 1500 with a resolution of 60,000, using a collision-induced dissociation (CID) fragmentation method. The raw files were analyzed using Sequest HT in the Proteome Discoverer v1.4.1.14 software (Thermo Fisher Scientific, San Jose, CA, USA). In addition, static side chain modification of the cysteines (Carbamidomethylation), as well as two dynamic modifications, were considered: the oxidation in methionine and glutamine and asparagine deamidation. Sequest was searched with a fragment ion mass tolerance of 0.6 Da and a parent ion tolerance of 20 ppm. Scaffold (version Scaffold_5.1.2, Proteome Software Inc., Portland, OR, USA) was used to validate the MS/MS-based peptide and protein identifications. The protein sequences stored in the UniProt and the coding transcripts obtained from the transcriptome were used to identify the peptides and proteins in the venom. Peptide identifications were accepted if they achieved greater than 99.0% probability and contained at least two identified peptides. Protein probabilities were assigned using the Protein Prophet algorithm [70]. Proteins that contained similar peptides and could not be differentiated based on MS/MS analysis alone were grouped to satisfy the principles of parsimony.

## Figures and Tables

**Figure 1 toxins-15-00498-f001:**
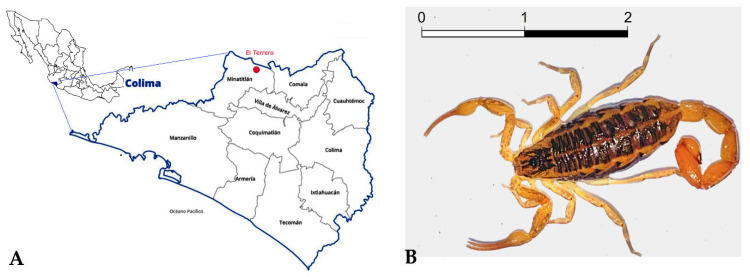
*Distribution and morphology of the scorpion Centruroides possanii*. (**A**) The orange dot indicates the geographical distribution and collection area of the scorpion *C. possanii* in Minatitlan, Colima, Mexico. (**B**) A female specimen of scorpion *C. possanii*. The scale bar 1:1 represents that 1 cm in the image equals 1 cm in reality.

**Figure 2 toxins-15-00498-f002:**
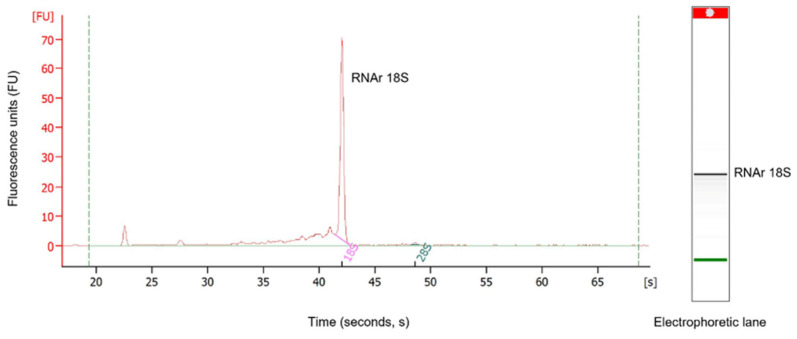
*RNA analysis using Bioanalyzer 2100 (Agilent).* The left image shows the representative electropherogram of the total RNA extracted from the venom-producing gland of the scorpion *C. possanii*. The separation of the 18S rRNA fragment in the gel is shown in the right image. The dotter points indicated in green refer to the internal standard of the equipment.

**Figure 3 toxins-15-00498-f003:**
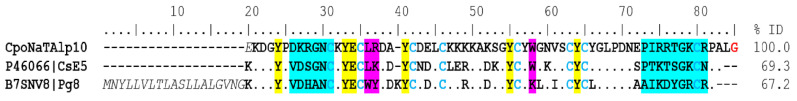
*Alignment of some alpha toxins that act on sodium channels.* The signal peptide is shown in gray and italics, and the mature peptide is in bold. Dots indicate identical amino acids. In blue is the identical pattern of the eight cysteines forming the disulfide bridges (positions 31–83, 35–56, 42–66, and 46–68). In yellow are the hydrophobic residues essential for their function on sodium channels; in cyan the variable region of the domain functional amino-carboxyl “NC”; and in pink, the amino acids of the functional “core” domain. Percent identity (% ID) is shown for mature peptides only. The amidation signal is represented in red color (position 85).

**Figure 4 toxins-15-00498-f004:**
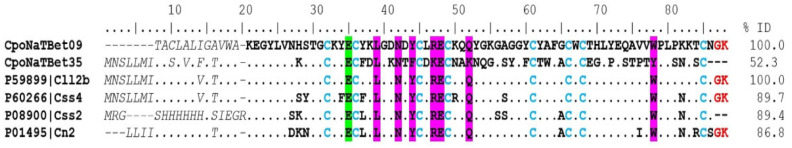
*Alignment of beta toxins that act on sodium channels.* The signal peptide shown is in italics and a gray color, and the mature peptide is in bold. Dots indicate identical amino acids. The identity percentage (% ID) refers only to the region of mature peptides. Cysteines (blue) present a conserved pattern formed by disulfide bridges at positions 31–84, 35–60, 44–65, and 48–67. Glutamic acid of position 35 (E35; green) is essential for binding β-NaTxs with the Nav channel. The crucial residues for the activity are shown in magenta, and the amidation and cut signals are in red.

**Figure 5 toxins-15-00498-f005:**
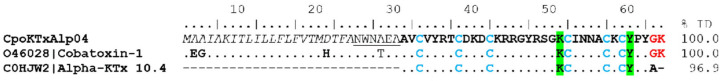
*Alignment of alpha toxins that act on potassium ion channels.* The signal peptide is shown in gray and italics. The underlined region represents the propeptide, the mature peptide is shown in bold, and in blue are the cysteines that form the disulfide bridges (positions 31–50, 35–55, and 40–57). Dots indicate identical amino acids. Amidation (G) and cleavage (K) signals are shown in red. The identity percentage (% ID) takes into account only mature peptides. The green shaded regions indicate the amino acids that form the functional dyad (K and Y) and interact with the potassium channel.

**Figure 6 toxins-15-00498-f006:**
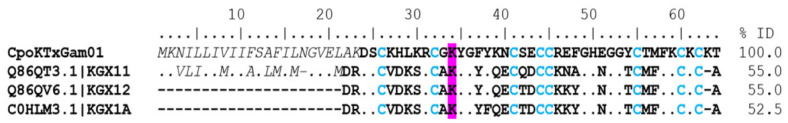
*Alignment of gamma toxins acting on potassium ion channels.* The signal peptide is shown in a gray color and italics. The mature peptide is shown in bold, and the cysteines forming the disulfide bridges are blue. Dots indicate identical amino acids. Highlighted in magenta is the amino acid involved with the binding site of the channel. The identity percentage (% ID) was estimated only for mature peptides.

**Figure 7 toxins-15-00498-f007:**
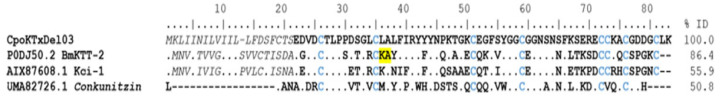
*Alignment of some group 1 delta toxins acting on potassium channels.* The signal peptide is shown in gray and italics, the mature peptide in bold, and the cysteines forming the disulfide bridges in blue. Dots indicate identical amino acids. In yellow, the amino acids Lys^36^ and Ala^37^ of the BmKTT-2 toxin are essential for function. The identity percentage (% ID) took into consideration only mature peptides.

**Figure 8 toxins-15-00498-f008:**
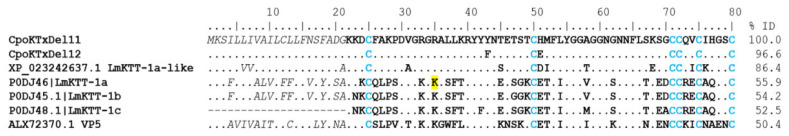
*Alignment of group 2 delta toxins acting on potassium ion channels.* The signal peptide is shown in gray and italics, the mature peptide in bold, and the cysteines forming the disulfide bridges in blue. Dots indicate identical amino acids. In yellow is the amino acid Lys^35^ of the LmKTT-1a toxin, which interacts directly with trypsin. The identity percentage (% ID) considered only the region of mature peptides.

**Figure 9 toxins-15-00498-f009:**
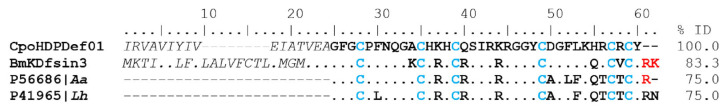
*Alignment of Host Defense Peptides of the Defensin type.* In italics are the amino acids of the signal peptide, in bold are the mature peptides, and dots indicate identical amino acids. The conserved cysteine pattern is in blue, and the amino acids indicate cleavage (RK) in red. The percentages of identity (% ID) between sequences correspond to the mature peptide.

**Table 1 toxins-15-00498-t001:** Scorpion transcriptome assembly and annotation for *C. possanii*.

Entry	Quantity
Sequence length	76 base pairs (bp)
Reads	19,158,736
Total transcripts	835,204
Annotated transcripts	720,463
Putative transcripts	598,252
Venom transcripts	434,492
No venom transcripts	163,760
Pfam annotation	28,399
Pfam scorpion domain	56
Pfam scorpion venom domain	50
GO terms	18.4% cell components 30.8% biological process 50.7% molecular function

**Table 2 toxins-15-00498-t002:** Results in summary of the *C. possanii* scorpion transcriptome.

Family	Subfamily	Number of Coding Transcripts
Toxins	α-NaTx	21
	β-NaTx	50
	α-KTx	20
	β-KTx	1
	γ-KTx	5
	δ-KTx	12
Enzymes	PA1	4
	PA2	4
	PLB	1
	MtP	16
	SeP	14
	Hya	1
	CiP	2
	PDE	3
	Mon	2
Inhibitory peptides	SERPIN	5
	TIL	11
	Kunitz type	1
Defense peptides	Defensin	10
	NDBP-3	1
Other components	CAP	7
	IGFBP	27
	La1-like	7
	Und	19
Total		244

Abbreviations: α-NaTx = alpha-type sodium toxins; β-NaTx = beta-type sodium toxins; α-KTx = alpha-type potassium toxins; β-KTx = beta-type potassium toxins; γ-KTx = gamma-type potassium toxins; δ-KTx = delta-type potassium toxins; PA1 = phospholipase type 1; PA2 = phospholipase type 2; PLB = phospholipase type B; MtP = metalloproteinase; SeP = serine protease; Hya = hyaluronidase; CiP = cysteine protease; PDE = phosphodiesterase; Mon = monooxygenase; SERPIN = serine protease inhibitor; TIL = trypsin inhibitor; Kunitz type = kunitz-type inhibitor peptide; NDBP-3 = non-disulfide-bridged peptide type 3; CAP = cysteine-rich secretory proteins, antigen 5, and pathogenesis-related 1 proteins (CAP superfamily); IGFBP = insulin-like growth factor-binding proteins; Und = undefined proteins.

**Table 3 toxins-15-00498-t003:** Results and summary of the *C. possanii* scorpion proteome.

Family	Subfamily	Number of Peptides and Proteins
Toxins	α-NaTx	4
	β-NaTx	14
	α-KTx	3
	β-KTx	1
	γ-KTx	1
Enzymes	PA1	3
	PA2	1
	PLB	1
	MtP	11
	SeP	4
	Hya	1
	CiP	1
	PDE	3
	Mon	2
Inhibitory peptides	SERPIN	4
	TIL	2
Defense peptides	Defensin	2
	NDBP-3	1
Other components	CAP	4
	IGFBP	1
	La1-like	1
	Und	5
Total		70

## Data Availability

A total of 19,158,736 reads were sequenced and are available in the European Nucleotide Archive (ENA), under accession number PRJEB60470.

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
