# Peer review of "Unveiling the Protein Components of the Secretory-Venom Gland and Venom of the Scorpion Centruroides possanii (Buthidae) through Omic Technologies"

_toxins, 2023, doi:10.3390/toxins15080498_

Round 1

Reviewer 1 Report

The manuscript reports findings of the secretory gland transcriptome and venom proteome of C. possanii scorpion from Mexico through omics technologies. The finding is relevant to the field and novel considering that data is lacking for this species which has potential medical importance in terms of scorpion sting intoxication and drug discovery. The study is well planned and the manuscript is well organized. However, there are some queries which deserve attention from the authors, and addressing these should help to improve the manuscript before it is considered for publication. The comments and suggestions are given below. 

Line 53-59: These are redundant to results and finding summarized in the abstract. The introduction is a little bit too short. Suggest to include the biological/medical importance and relevance of scorpion stings in this region (especially where this species or genus of scorpion is relevant). For instance, the epidemiology and clinical effect. This will justify the need for and objective of the study.

Figure 1. A - geographical distribution is not clear. Please indicate with shading or other indicators to mark the geographical distribution of the species. In addition, the description for the orange dot should belong to panel A, not B. 

Line 68-70: The sentence contains grammatical errors. Please correct it. 

Line 60-71: Authors claimed the signs observed are "certainly related to neurotoxins present in ...."-- this is only a hypothesis and suggest authors not to use words like "certainly related to...". 

Line 72: "...described for other scorpions of the same genus." -- References are needed here. 

Line 76: Authors mentioned "RIN number" - Is there a RIN number for this sample?

Line 109: ....the most abundant (?). 

Table 2 and Figure 3: Authors presented the findings and reported "abundance relative to the annotated transcripts putatively coding for venom components of the scorpion". It is not clear how the "abundance" is referring to and how it is determined. Was it based on the ratio of number of coding transcripts identified, or more accurately shouldn't it be analyzed based on the expression levels using parameters such as RPKM, FPKM etc.? The relative abundance of transcripts based on gene expression profiling is usually represented in percentage in most venom gland transcriptomic studies. Please clarify. 

Also, please include footnotes for all abbreviations (for tables and figures). 

Line 121-122: "previous reports of...." -- authors should provide the references of these "previous reports". 

Line 128-129" "....toxic to mammalians and insects." - Please elaborate more on the "toxicity" and included the reference(s) here. 

Line 171: Please explain how "complete sequences" are determined? 

Line 262-272 and throughout the text: The "2" in PLA2 should be subscript. 

Line 311-312: it is interesting to note the presence of conserved metallopeptidase motifs in the proteases. Does the metallopeptidase (of the related scorpions) known to cause local effect such as bleeding? 

Line 478 and throughout the text: Please check all genus and species names for italic style. 

Section 2.5.2. Proteomics - Authors included LCMS/MS study for the soluble venom, adding values to the transcriptomic study. Is there any result of relative abundance of the protein/peptide identified, and is there any correlation between the transcriptome and the proteome? Suggest to discuss. 

Line 515 - Please explain the "RNAse-free conditions", how these were conducted. 

Line 555: Does the venom sample include the venom for the specimens used in transcriptomics? 

Line 605-606: Is there any ethics approval code for this authorization mentioned here? Suggest to state here.

Language and grammars need to be checked. 

Author Response

Dear Reviewer 1,

Thank you for allowing us to correct the submitted manuscript Toxins-2492799. We made some changes to our manuscript, according to your suggestions, hoping to find a better version and that our changes have answered all your recommendations.

Line 53-59: These are redundant to results and finding summarized in the abstract. The introduction is a little bit too short. Suggest to include the biological/medical importance and relevance of scorpion stings in this region (especially where this species or genus of scorpion is relevant). For instance, the epidemiology and clinical effect. This will justify the need for and objective of the study.

Answer: Thanks, the paragraph was corrected.

Figure 1. A - geographical distribution is not clear. Please indicate with shading or other indicators to mark the geographical distribution of the species. In addition, the description for the orange dot should belong to panel A, not B.

Answer: The scorpion C. possanii only has been found in El Terrero. Its geographical distribution is indicated in the orange dot in the map of Figure 1A.

Line 68-70: The sentence contains grammatical errors. Please correct it.

Answer: The sentence was corrected. Lines 76-83.

Line 60-71: Authors claimed the signs observed are "certainly related to neurotoxins present in ...."-- this is only a hypothesis and suggest authors not to use words like "certainly related to...".

Answer: The sentence was corrected. Lines 80.

Line 72: "...described for other scorpions of the same genus." -- References are needed here. Line

Answer: The references was added. Lines 81-83.

Line76: Authors mentioned "RIN number" - Is there a RIN number for this sample?

Answer: In scorpions, the RIN number cannot be measured because the 28S rRNA subunit when heated, gives two segments that migrate together with the 18S (see reference Natsidis et al., 2019), as mentioned in this phrase at line 87.

Natsidis, P.; Schiffer, P.H.; Salvador-Martínez, I.; Telford, M.J. Computational Discovery of Hidden Breaks in 28S Ribosomal RNAs across Eukaryotes and Consequences for RNA Integrity Numbers. Sci. Rep. 2019, 9, 19477, doi:10.1038/s41598-019-55573-1.

Line 109: ....the most abundant (?).

Answer: The sentence was corrected. Line 117.

Table 2 and Figure 3: Authors presented the findings and reported "abundance relative to the annotated transcripts putatively coding for venom components of the scorpion". It is not clear how the "abundance" is referring to and how it is determined. Was it based on the ratio of number of coding transcripts identified, or more accurately shouldn't it be analyzed based on the expression levels using parameters such as RPKM, FPKM etc.? The relative abundance of transcripts based on gene expression profiling is usually represented in percentage in most venom gland transcriptomic studies. Please clarify.

Also, please include footnotes for all abbreviations (for tables and figures).

Answer: Figure 3 was removed. The TMP values indicate the relative abundance of the transcripts, shown in Supplementary Table 1. Also, all abbreviations were added in Table 2 (lines 121-130).

Line 121-122: "previous reports of...." -- authors should provide the references of these "previous reports".

Answer: Reference was added in the line 141.

Line 128-129" "....toxic to mammalians and insects." - Please elaborate more on the "toxicity" and included the reference(s)

Answer: This was corrected in lines 146-152.

Line 171: Please explain how "complete sequences" are determined?

Answer: Complete sequences refer to coding transcripts containing all amino acids corresponding to the mature peptide. This is mentioned in section 4.4, in lines 691-693.

Line 262-272 and throughout the text: The "2" in PLA2 should be subscript.

Answer: The two number is correct.

Line 311-312: it is interesting to note the presence of conserved metallopeptidase motifs in the proteases. Does the metallopeptidase (of the related scorpions) known to cause local effect such as bleeding?

Answer: Metalloproteases can inhibit platelet aggregation affecting coagulation. However, have been known for more than 35 years that venom scorpions affect mammalian blood coagulation. This has been reported in envenoming by Mesobuthus tamulus (Devi et al., 1970; Reddy et al., 1972; Sarka et al., 2008), M. martensii (Song et al., 2005), Centruroides sculpturatus (Longenecker and Longenecker, 1981), Palamneus gravimanus, Leiurus quinquestriatus (Hamilton et al., 1974), Tityus serrulatus (Souza et al., 2013), T. discrepans (D'Suze et al., 2001, 2003), and Hemiscorpius lepturus (Khataminia et al., 2020), have all shown the ability to disturb the hemostatic system. However, the components responsible for these activities and their action mechanisms are unknown.

Line 478 and throughout the text: Please check all genus and species names for italic style. Section 2.5.2. Proteomics - Authors included LCMS/MS study for the soluble venom, adding values to the transcriptomic study. Is there any result of relative abundance of the protein/peptide identified, and is there any correlation between the transcriptome and the proteome? Suggest to discuss.

Answer: The text was corrected, and Discussion was added in section 2.6.

Line 515 - Please explain the "RNAse-free conditions", how these were conducted.

Answer: This work used the following conditions throughout the experiment: to use disposable gloves and change them frequently. Glassware and plastic ware RNase free were used. Scalpels, tweezers, and forceps were washed with detergent and baked at 210°C for four hours before. Also, reagents that are certified to RNase free were used for RNA analysis.

Line 555: Does the venom sample include the venom for the specimens used in transcriptomics?

Answer: Yes, I do. The venom was obtained from the same scorpions used in transcriptomic analysis. The venom extraction was performed first and after the RNA extraction.

Line 605-606: Is there any ethics approval code for this authorization mentioned here? Suggest

to state here.

Answer: Ethics approval code was added, line 575-576.

Language and grammars need to be checked.

Answer: Language and grammars was corrected.

Reviewer 2 Report

The application of omics technology has revolutionized the identification of individual components in complex samples, such as toxins and peptides in animal venoms. The present study illustrates the application and richness of useful information provided through an omics exploration of venoms. The molecular and functional diversity of toxins and peptides has been the basis of basic and applied investigations and the development of new drugs or therapies for accidents involving venomous animals. Thus, the present study expands the biochemical knowledge of the composition of scorpion venoms, through the study of the composition of the venom of Centruroides Possonii (Buthidae) through omic approaches.

1. The abstract section must contain the main findings of the manuscript. Lethality results were omitted.

2. The key contribution statement does not describe the potential implications of the presented results. In the current version of the manuscript, this section just repeats the results without highlighting the important contribution of this study.

3. The authors describe the results in the last section of the introduction, which makes results and discussion repetitive. I recommend only presenting the state of the art and research objectives in the introduction.

4. The quality of Figure 1A needs to be improved.

5. Please include a scale bar in Figure 1B.

6. Express the lethal dose as ug/g.

7. Express the lethal dose as ug/g. Additionally, what deviation is the range. Was this experiment performed only once?  Compare the LD50 with other species.

8. This species should be considered of medical importance. What is the criteria for this suggestion?

9.  Table 2. Please include the meaning of acronyms in the table footer.

10. Figure 11 could be presented as supplementary material. Authors should keep only the main findings in the article.

11. The authors do not discuss the differences and reasons behind the differences between transcriptomics and proteomics results. Additionally, they mention that this study presents the largest number of characterized components of scorpion venom using omics technologies. However, it would be interesting to compare with previous studies and explain what allowed this greater identification.

12. The order of description of the methodology should coincide with the presentation of results. For example, the authors describe the lethality results prior to the identification of venom components.

13. The dose described as LD50 does not match the animals' weight described in the methodology. From the expression of the dose, it seems that they used 20 g animals, however, the methodology shows a different weight.

14. How long were venom-injected animals monitored in the lethality assay? It would be interesting to check a survival curve alongside time for the doses tested.

15. What doses were used in the lethality assay?

16. Authors must include details of approval of animal experimentation by the institutional ethics committee (number license or project).

17. The implications and relevance of the presented results is not clearly described in the work. The implications and relevance of the presented results is not clearly described in the work. The conclusion lacks perspective.

18. The work is limited to venom component identification. The functional view was not explored. In this sense, the authors must clarify the limitations of the study and which steps are important from the information generated.  

Author Response

Reviewer #2: The application of omics technology has revolutionized the identification of individual components in complex samples, such as toxins and peptides in animal venoms. The present study illustrates the application and richness of useful information provided through an omics exploration of venoms. The molecular and functional diversity of toxins and peptides has been the basis of basic and applied investigations and the development of new drugs or therapies for accidents involving venomous animals. Thus, the present study expands the biochemical knowledge of the composition of scorpion venoms, through the study of the composition of the venom of Centruroides posanii (Buthidae) through omic approaches.

Dear Reviewer 2,

Thank you for allowing us to correct the submitted manuscript Toxins_2492799. We made some changes to our manuscript, according to your suggestions, hoping to find a better version and that our changes have answered all your recommendations.

  1. The abstract section must contain the main findings of the manuscript. Lethality results were omitted.

Answer: Lethality was added in abstract section, lines 26-27.

  1. The key contribution statement does not describe the potential implications of the presented results. In the current version of the manuscript, this section just repeats the results without highlighting the important contribution of this study.

Answer: The key contribution was added, lines 36-40.

  1. The authors describe the results in the last section of the introduction, which makes results and discussion repetitive. I recommend only presenting the state of the art and research objectives in the introduction.

Answer: This section was corrected.

  1. The quality of Figure 1A needs to be improved.

Answer: The quality of the Figure 1A was corrected.

  1. Please include a scale bar in Figure 1B.

Answer: The scale bar in Figure was added.

  1. Express the lethal dose as ug/g.

Answer: The lethal dose was added, line 77.

  1. Express the lethal dose as ug/g. Additionally, what deviation is the range. Was this experiment performed only once? Compare the LD50 with other species.

Answer: This was added in the manuscript in lines 76-77.

  1. This species should be considered of medical importance. What is the criteria for this suggestion?

Answer: This specie should be considered of medical importance by its LD50 value and by the intoxication signs observed could be related to neurotoxins present in C. possanii venom, which affect excitable cells, as described for other scorpions of the same genus like C. tecomanus (10.2 μg/20 g mouse) and C. limpidus (15.0 μg/20 g mouse). This is included in the text, lines 78-83.

  1. Table 2. Please include the meaning of acronyms in the table footer.

Answer: This Table was removed of manuscript.

  1. Figure 11 could be presented as supplementary material. Authors should keep only the main findings in the article.

Answer: This Figure was removed, and the result was included in the manuscript, lines 486-491.

  1. The authors do not discuss the differences and reasons behind the differences between transcriptomics and proteomics results. Additionally, they mention that this study presents the largest number of characterized components of scorpion venom using omics technologies. However, it would be interesting to compare with previous studies and explain what allowed this greater identification.

Answer: This was added in section 2.6.

  1. The order of description of the methodology should coincide with the presentation of results. For example, the authors describe the lethality results prior to the identification of venom components.

Answer: Thank you for your suggestion. The lethality results were left in this order.

  1. The dose described as LD50 does not match the animals' weight described in the methodology. From the expression of the dose, it seems that they used 20 g animals, however, the methodology shows a different weight.

Answer: The assays were realized in mice of 25 – 30 g. The dose of venom injected was just based on mouse weight, which permits determining the LD50 in ug / 20 g of mouse weight.

  1. How long were venom-injected animals monitored in the lethality assay? It would be interesting to check a survival curve alongside time for the doses tested.

Answer: The venom-injected mice were monitored at 24 h. After this time, the mouse was considered alive or dead. This was added to the text in line 574.

  1. What doses were used in the lethality assay?

Answer: This was added to the text, in lines 571-572.

  1. Authors must include details of approval of animal experimentation by the institutional ethics committee (number license or project).

Answer: This was added to the text, in line 576.

  1. The implications and relevance of the presented results is not clearly described in the work.. The conclusion lacks perspective.

Answer: This was added to the text, in lines 541-554.

  1. The work is limited to venom component identification. The functional view was not explored. In this sense, the authors must clarify the limitations of the study and which steps are important from the information generated.

Answer: This was added to the text, in lines 541-554.

Reviewer 3 Report

My comments to the authors include many examples of grammatical and syntax issues. I believe once these problems are shown to occur systematically throughout the manuscript the reviewer has done their duty. Systematic language issues need to be addressed and English language editing by a professional editor would greatly improve this manuscript. 

Author Response

Reviewer 3

Manuscript ID: toxins-2492799

Dear Reviewer 3,

Thank you very much for the favorable commentaries and congratulations for the work we submitted to TOXINS for publication.

I recognize the efforts and extended revision you performed on our manuscript, including the work of suggesting a new format for several of our initially proposed text-style.

We have made most of the corrections and modified or responded to your suggestions. We thank you very much; we greatly appreciate it.

A small number of your suggestions or questions were not properly responded to, because as you recognized, you are unaware of the details involved in this type of work. However, people working with transcriptome analysis will certainly understand what we wrote.

I will comment some of your questions and recommendations.

  1. Two important points were taken into consideration, the first was to make a complete revision of the abstract (summary), and the second was to clarify the duplicate information contained in Table 2 and Fig.3. We accepted your suggestion and eliminate Fig.3.
  2. Concerning your commentaries number 3, 4, and 5.

We preferred to leave it in the order that it is described because probably the most important new information reported refers to the transcriptome results. But you are right; the bioassay (LD50 estimation) must be included in the Summary section, which we did. As requested, we also reported the nanograms/gram mouse weight values. English language was revised, as much as we could, and corrected several errors.

We understand your request of defining some words before giving the results found, but unfortunately, the journal TOXINS instruction to authors requires that the Material and Methods be placed at the end of the Results and Discussion section. As mentioned, experts in transcriptomic analysis will understand the words and abbreviations used regarding the procedures used for describing the results presented.

  1. Concerning your points 6 and 7.

You are right; we emphasized some findings and others not. We have chosen to call attention to some of our findings that we thought were more relevant and showed the originality of the work, compared to other similar publications in the literature. However, we want to stress that this is not a review article, and we cannot dedicate so much effort to discussing each of the results we describe in the same manner. I hope you understand this point! Concerning milking the scorpions for obtaining the venom, our protocols are very well known in described in the literature. Similarly, the “Up and Down” method is well known. It is the best method for saving animals, avoiding the submission of toxic materials to living organisms, especially vertebrates. The permit given to us for this type of assay is very restrictive. We cannot use many animals. Here we used six females and six males. Finally, your question about waiting two weeks to use the venomous gland of the scorpions is also well-known by our group (please see: Carcamo-Noriega, E. N., Possani, L. D., & Ortiz, E. (2019). Venom content and toxicity regeneration after venom gland depletion by electrostimulation in the scorpion Centruroides limpidus. Toxicon, 157, 87-92. doi:10.1016/j.toxicon.2018.11.305). The scorpions need at least two weeks to regenerate the content of proteins in their glands. However, you are right “the regeneration of specific venom components is asynchronous,” but this has not been described in detail yet.

  1. Specific points referred line by lines.

Most of the points were corrected, as suggested. We have just a few commentaries.

“Line 7 and elsewhere: mammalians should be replaced with mammals. This change should be made throughout the entire manuscript”. Both terms are equivalent and can be used indistinctly any time, in our opinion.

“Lines 20-21: It may be redundant to include words that appear in the title in the list of keywords too. I recommend removing words that appear in the title from the keywords”.

We are not sure of this. When you use a word as keyword, you do not know what the title of the article is. In our opinion, this is not redundant, but it helps finding the proper title of the manuscript under query.

“Line 22 (and elsewhere): the phrase “venomous gland” should be replaced with “venom gland.” We do not agree with this recommendation because, grammatically, it is correct. Venom is a substantive, gland is also a substantive, whereas venomous is an adjective that qualifies the following substantive. The gland is not any kind of gland, it is a specialized gland that produces venom. We know people use venom gland instead of venomous gland, but in our opinion, it is grammatically wrong.

“Lines 79-81: the sentence “Nonetheless, the clearly defined 18S rRNA band in the gel and its peak in the electropherogram is observed, indicating RNA quality and integrity.” I would benefit from references that support this claim”. The photo on the right side of Fig.2 is the gel given by the Bioanalyzer 2100 of Agilent.

“Figure 1: Part A of Figure 1 says that the map depicts the Geographic distribution of C possanii. Is this true? This is a new species and may have only been observed in a single location so far but is it true that this single reserve is the only place this scorpion is found? Maybe it would be better to say that this is the location where these scorpions were collected instead of its geographic range”. The response to your first question is YES, and the second WE DO NOT KNOW for sure, but since this is a special mountainside, separated from the surrounding area, we assume it is the only place thus far visited, where this scorpion is found.

“Lines 53-54 up to the phrase starting with Table2.

Most of the observations were taken into consideration and corrected accordingly, except words that are well known by people in the field.

“Table 2: I think the information in Table 2 could easily be added to Figure 3. Much of the information is already present; thus, I recommend consolidating these into one and removing Table 2”. You are right, we eliminate Figure 3 and corrected the text accordingly.

“Line 124: change “surmised” to “were assumed”. In my opinion is not necessary, because both terms are correct and in my opinion “surmised” is more elegant.

“Lines 126-127 up to Line 271”. We made modification in most of the cases, as requested.

“Line 271: I have not heard PLA2 be referred to as spreading factors before. I am more familiar with hyaluronidases being referred to as spreading agents. Please double-check this and correct it if necessary”. You are right, most people know that the enzyme hyaluronidase is considered to be a spreading factor, however, since PLA2 hydrolyzes phospholipids, which are important components for stabilization of membranes; if this is modified by PLA2 it makes the cells more accessible to incoming factors. In our opinion it is correct to call them spreading factors. The correction was done.

“Line 281 to Line 341”. The corrections were done.

“Lines 350-351: I am confused by the sentence “This is the first report demonstrating the presence of phosphodiesterases in scorpion venoms.” Is this referring to this study or one of the studies cited above? I am assuming that this is referring to this manuscript, but how does that fit with the sequence identity to “phosphodiesterase-type sphingomyelin (NCBI XP_023216777.1) from the scorpion C. sculpturatus”?

Please be aware that the phosphodiesterase found in C. scupturatus was done only in the trancriptome results. Here we are reporting for the first time the presence of this enzyme in the venom of C. possanii.

“Lines 354-355 up to end” We corrected most of it, accordingly.

“Line 761 up to Supplemental Table 3”

The corrections were done.

  1. The extra comments that you made were not taken into consideration, because the obligation of the reviewers is not to share the information contained in the manuscript submitted for publication, until they are accepted by the editor.

However, as you can see, the three reviewers that evaluate this manuscript have manifested favorable commentaries, thus I ask you please tell the Editor that you accept my responses as adequate. Thank you

Round 2

Reviewer 2 Report

Authors have addressed most of previous comments. However, there are still pending points to be clarified. 

1. The parameters for classification of this species as medically important are not enough. Toxicity and venom composition not always translate (mean) in human envenomation. Epidemiologic aspects, distribution and other points should be taking account here. Thus, authors should revise this statement. 

2.  The determination of the LD50 is unclear. The authors do not specify how many males and females. They used 6 for find the concentration? How was this performed? One concentration per mouse? How many concentrations was tested? What is the confidence interval? How to determine it using the approach described.  

3. Authors should explain how they calculate this values using the methodology described in section 2.1. 

4. Figure 1. Does the orange dot actually represent distribution of this species?  Authors should show the distribution in a different color compared to the collection site. What was the biodiversity database used to represent the distribution?

Author Response

29 June 2023

Date of this review

25 July 2023 19:40

Dear Reviewer 2

Thank you for allowing us to correct the submitted manuscript Toxins_2492799. We made some changes to our manuscript, according to your suggestions, hoping to find a better version and that our changes have answered all your recommendations.

Reviewer 2: Authors have addressed most of previous comments. However, there are still pending points to be clarified.

  1. The parameters for classification of this species as medically important are not enough. Toxicity and venom composition not always translate (mean) in human envenomation. Epidemiologic aspects, distribution and other points should be taking account here. Thus,

authors should revise this statement.

Answer. In Mexico, scorpions of medical importance belong to the genus Centruroides, known as "striped ones" (González-Santillán & Possani, 2018). The study to determine the median lethal dose (LD50) made it possible to determine the signs of direct intoxication of the C. possanii scorpion venom in mice of the Balb/c strain. As expressed in the manuscript (lines 78 and 79), the mice presented clear signs of poisoning such as spasms, spontaneous jumping, tachypnea, and tachycardia, which is considered the first reason for their consideration as a species of medical importance. Secondly, the LD50 obtained (13.18 ± 0.67 μg/20 g mouse) is similar to that of other scorpion species that inhabit Mexico and that are also considered to be of medical importance (lines 81-83 of the manuscript, Ponce-Saavedra et al., 2016; Riaño-Umbarila et al., 2017). Finally, the morbidity in the state of Colima, caused by five species of scorpions of the Centruroides genus, including C. possanii, was 9,015 cases reported in 2022, positioning itself as the fourth state in Mexico with the highest incidence of poisoning caused by scorpion sting, according to the Ministry of Health (Secretaría de Salud, 2022).

With the previous data, it is evident that the scorpion C. possanii presents the taxonomic, toxicological (experimental) and epidemiological criteria to be considered as a species of scorpion of medical importance.

  1. The determination of the LD50 is unclear. The authors do not specify how many males and females. They used 6 for find the concentration? How was this performed? One concentration per mouse? How many concentrations was tested? What is the confidence interval? How to determine it using the approach described.
  2. Authors should explain how they calculate this values using the methodology described in section 2.1.

Answer. Twelve male mice of the Balb/c strain were used to determine the LD50: six to obtain the dose and six for confirmation. The venom stock used for the test presented an absorbance of 0.608 g in 1 ml of soluble venom. The dose injected into the first mouse was 15 μg of venom for every 20 g of mouse weight. The first test was carried out as follows:

Mouse 1 weight = 28.6 g

Amount of venom to be administered: 15 μg/ 20 g of mouse weight

Calculation:

15 μg - 20 g mouse

                    ? - 28.6 g mouse

                              ? = 21.45 μg of venom

If there is 0.608 g of venom in 1 ml, then:

0.608g - 1ml

21.45 μg - ?

? = 35.27µl

In order to inject the dose of 15 μg of venom for every 20 g of mouse weight, mouse 1 was injected with 35.27 µl of the venom stock, volumetric to 50 ml with physiological saline (PBS). In the same way, the dose to the weight of each mouse.

According to the proposed by Dixon (1965), if the first mouse dies with the administered dose (15 μg of venom), the second mouse should be administered half the dose (7.5 μg of venom); if mouse 2 dies, mouse 3 is given half the previous dose (3.75 µg of venom) but if mouse 2 lives, mouse 3 is given the first dose again (15 µg of venom) and so on successively until applying the venom to each mouse. If the mouse lives after 24 hours of injection, it is recorded with an O, if the mouse dies it is recorded with an X.

The doses administered and the record of the events obtained in the assay to obtain the LD50 of the C. possanii venom are indicated below:

Mouse 1: 15 μg of venom per 20 g of weight, event record: X

Mouse 2: 7.5 μg of venom per 20 g of weight, event record: O

Mouse 3: 15 μg of venom per 20 g of weight, event record: O

Mouse 4: 30 μg of venom per 20 g of weight, event record: X

Mouse 5: 15 μg of venom per 20 g of weight, event record: X

Mouse 6: 7.5 μg of venom per 20 g of weight, event record: O

Pattern: XOOXXO

Making use of the previous pattern and table 1 published in the article by Dixon (1965), calculations were made to obtain the LD50:

Formula: Xf + kd

Where: Xf is the logarithm of the difference between the last and penultimate doses, k is the tabular value according to the pattern obtained and d is the interval between each dose.

Calculation:

Penultimate dose: 15 μg

Last dose: 7.5 μg

Difference between doses: 7.5 μg

Logarithm of 7.5 = 8.875

Table value = 0.831

Interval between doses = 0.295

Substitution:

0.875 + (0.831) (0.295)

= 0.245145 + 0.875

= 1.120145

Antilog of the previous value = 13.18

LD50 = 13.18 μg/20 g mouse

To validate the resulting dose, six male mice were injected at the same time with the obtained LD50. After 24 hours, half of the mice died.

  1. Figure 1. Does the orange dot actually represent distribution of this species? Authors should show the distribution in a different color compared to the collection site. What was the biodiversity database used to represent the distribution?

Answer: The scorpion C. possanii only has been found in El Terrero. Its geographical distribution as well as its collection site is indicated in the orange dot in the map of Figure 1A.

Round 3

Reviewer 2 Report

I have no further comments. The manuscript has been update accordingly.